# The Sweet Side of IVF: Biological Role and Diagnostic Potential of Galectin-9 in Female Infertility

**DOI:** 10.3390/ijms26167672

**Published:** 2025-08-08

**Authors:** Beata Polgar, Matyas Meggyes, Krisztina Godony, Akos Varnagy, Kalman Kovacs, Peter Mauchart, Peter Matrai, Krisztina Kovacs, David Semjen, Tamas Tornoczki, Laszlo Szereday

**Affiliations:** 1Department of Medical Microbiology, Medical School, University of Pecs, 12 Szigeti Street, 7624 Pecs, Hungary; meggyes.matyas@pte.hu (M.M.); szereday.laszlo@pte.hu (L.S.); 2Janos Szentagothai Research Centre, University of Pecs, 20 Ifjúság Street, 7624 Pecs, Hungary; 3Department of Obstetrics and Gynecology, Medical School, University of Pecs, 17 Édesanyák Street, 7624 Pecs, Hungary; krisztina.godony@cdoki.hu (K.G.); varnagy.akos@pte.hu (A.V.); kovacs.kalman@pte.hu (K.K.); mauchart.peter@pte.hu (P.M.); 4MTA-PTE Human Reproduction Scientific Research Group, Human Academy of Sciences, 17 Édesanyák Street, 7624 Pecs, Hungary; 5Institute of Bioanalysis, Medical School, University of Pecs, 12 Szigeti Street, 7624 Pecs, Hungary; peter.matrai@pte.hu; 6Department of Pathology, Medical School and Clinical Centre, University of Pecs, 12 Szigeti Street, 7624 Pecs, Hungary; kovacs.krisztina2@pte.hu (K.K.); semjen.david@pte.hu (D.S.); tornoczki.tamas@pte.hu (T.T.)

**Keywords:** infertility, follicular fluid, Galectin-9, biomarker, in vitro fertilization

## Abstract

Infertility rates are indeed increasing globally, which emphasizes a pressing need to identify novel biomarkers exhibiting superior potential for laboratory diagnosis and personalized clinical management. This study aimed to explore the biological role of Galectin-9 (Gal-9) in female fertility and evaluate its diagnostic potential in the In Vitro Fertilization (IVF) program. A prospective cohort study was performed on 83 follicular fluids (FF) and 19 serum-FF pairs from IVF patients, 16 serum samples from fertile women, and 12 tissue sections. Gal-9 expression was characterized by immunostaining and ELISA. The ROC analysis was employed to evaluate the overall diagnostic performance. Cell-specific ovarian Gal-9 expression and significant differences in soluble Gal-9 levels were identified in the serum and FF of fertile and infertile women. Elevated intrafollicular Gal-9 levels were linked to poor ovarian reserve, served as a predictive marker for ovarian hyperstimulation, and marked unfavorable IVF outcomes. Follicular Gal-9 levels positively correlated with peak estradiol and total daily FSH dosage. ROC analysis revealed an excellent diagnostic value of Gal-9 for predicting fertilization success and a moderate ability to predict IVF outcomes. Our findings suggest a potential role for Gal-9 in oogenesis and its promise as a diagnostic marker for predicting fertilization success in IVF. However, further studies are needed to confirm its clinical utility in assisted reproduction.

## 1. Introduction

Infertility affects 17.5% of the reproductive-aged population worldwide [1], which is defined by the inability to conceive after at least 12 months of regular, unprotected intercourse. The etiology is multifaceted, including female (50%) or male factors (20–25%), combined factors (25–30%) [2], or unknown origin (11%) [3]. Female infertility often results from anatomical or functional abnormalities of the reproductive organs or endocrine system and is clinically classified as either primary or secondary, depending on the presence or absence of previous pregnancies. For couples unable to conceive naturally, in vitro fertilization (IVF) may provide the only opportunity to achieve a pregnancy [4].

The increasing rate of reproductive problems emphasizes the pressing need to aid the early diagnosis and individualized clinical management of infertility, thereby improving Assisted Reproductive Technology (ART) success rates. During the last decade, scientific investigations have facilitated the identification of potential fertility biomarkers from diverse biological samples. Among them, follicular fluid (FF) has become one of the most extensively studied biofluids in female fertility research due to its critical role in oocyte maturation and ovulation [5,6,7,8]. As the follicular microenvironment critically influences oocyte quality and pregnancy outcomes [9], alterations in follicular fluid composition may reflect ovarian dysfunction and predict ART-related complications, such as ovarian hyperstimulation syndrome (OHSS) [10,11,12,13,14].

Galectins (Gal), an evolutionarily conserved family of β-galactoside-binding lectins, control diverse cellular processes and participate in the regulation of complex cell-cell-matrix interactions, immunity, and tolerance [15]. Of the 19 identified mammalian galectins, 16 are expressed at the maternal-fetal interface, with Gal-1–4, -7–9, and -12 present in the normal endometrium [16,17,18,19]. This study focuses on Gal-9, a tandem-repeat-type lectin, crucial for female reproduction [20]. Initial reports indicated that Gal-9 is exclusively expressed in endometrial epithelial cells [21], suggesting its involvement as a marker of receptivity and its pivotal role during blastocyst implantation [22]. Subsequent investigations revealed that various *LGALS9* mRNA splice variants exhibit divergent functions during normal and pathological pregnancy [23]. Over the past 15 years, Gal-9 has been linked to maternal-fetal immune tolerance [24,25,26], placental development [27,28], and maternal vascular expansion [29]. Meanwhile, Gal-1, -3, -7, and -9 have been studied in fertility disorders [30,31,32,33]. Unfortunately, data on the diagnostic utility of Gal-9 in female infertility remains limited. Therefore, we aimed to investigate the role of Gal-9 in female infertility and reveal its potential as a minimally invasive diagnostic biomarker in clinical practice.

This study provides preliminary evidence that Gal-9 may contribute to oocyte development and could serve as an early biomarker for predicting fertilization success in IVF. However, validation in larger, multicenter cohorts is necessary to confirm its function during oogenesis and establish its diagnostic value in assisted reproductive technologies.

## 2. Results

### 2.1. Galectin-9 Expression in the Human Ovary

Despite extensive research on *LGALS9* mRNA expression in the endometrium and during pregnancy [23], its role in oocyte development remains unclear. To explore this effect, we used immunohistochemistry to investigate ovarian Gal-9 expression patterns in various stages of human follicle development. In line with the previous observation by Labrie et al. [34], weak Gal-9-specific immunostaining was observed in ovarian epithelial and stroma cells (Figure 1(1)). Besides, we detected weak to moderate labeling in primordial and primary follicles, mainly in the cytoplasm of the enclosed oocytes (Figure 1(2,3)). In the tertiary follicles, we found intense nuclear positivity in some of the theca interna (TI) and theca externa (TE) cells and reduced labeling in the mural granulosa (MG) (Figure 1(4,4a)). In Graafian follicles, the nuclei of some cumulus granulosa (CG) cells and the plasma membrane (PM) of the mature oocyte also exhibited positive labeling (Figure 1(4b)). The specificity of the Gal-9 immunostaining was confirmed on a secretory phase endometrium (positive control) and an ovary section stained with an isotype-matched non-specific monoclonal antibody (negative control) (Figure 1(5,6)). Figure 1(1′–4′) shows the hematoxylin and eosin (H&E) stained images of the same sections.

### 2.2. Association with IVF-Related Parameters

Following confirming Gal-9 expression in the ovary, we examined the soluble Gal-9 content of the aspirated follicle fluid of infertile patients. The intrafollicular concentration of soluble Gal-9 ranged from 101.23 to 446.01 pg/mL, with a decreasing trend observed in response to ovarian stimulation. The highest lectin levels were detected in low responders (<3 collected oocytes), while the lowest values were found in high responders (>15 collected oocytes). Surprisingly, two cases with significantly elevated (*p* = 0.0255) follicular Gal-9 values subsequently developed ovarian hyperstimulation syndrome (Figure 2A).

Next, we investigated the potential of intrafollicular Gal-9 levels in the prediction of fertilization success and IVF outcomes. Women with successful fertilization (SF) had significantly higher follicular Gal-9 levels (*p* = 0.0098) than those with unsuccessful fertilization (UF) (Figure 2B). Prospectively, when considering IVF outcomes, significantly elevated lectin levels (*p* = 0.0017) were found in the FF samples of the non-pregnant (NP) cohort in comparison to the group where the fertilization was unsuccessful and no embryo transfer (NoET) was performed (Figure 2C). Our detailed analysis suggests that changes in follicular Gal-9 levels may depend on both the number of stimulation cycles and the outcomes of IVF. While statistical significance was not detected, a notable pattern emerged. Specifically, in patients who achieved pregnancy following a fresh embryo transfer, the measured Gal-9 levels showed an overall downward trend across the three cycles of the entire IVF program. In contrast, in patients who did not conceive, lectin levels increased consistently from the first to the fifth stimulation cycle (Figure 2D).

Finally, since the duration of infertility can be a prognostic factor for the success rate of IVF [35], we investigated the potential impact of this predictor on soluble Gal-9 levels. We noticed that those women with infertility lasting over 5 years exhibited higher follicular lectin content compared to those under 5 years, albeit no statistically significant difference or correlation was observed (Figure 2E). Other clinical parameters, such as the main causes of infertility, the number of previous pregnancies, and earlier surgical interventions, had no significant effect on intrafollicular Gal-9 levels.

### 2.3. Association with Laboratory Parameters

As the final composition of FF significantly impacts oocyte maturation, fertilization, and embryo development [6,36], we examined the possible correlation between soluble, follicular Gal-9 and serum hormone levels essential for optimal oocyte development.

*Serum estradiol (E2):* In women with peak serum E2 levels exceeding 3000 pmol/L, we found a significant increase (*p* < 0.05) in follicular Gal-9 levels and a weak positive correlation between the examined factors (r_s_ = 0.257, *p* = 0.0249) (Figure 3A,B). When we separately analyzed the subgroup of women who did or did not conceive, an opposing trend emerged with a weak positive correlation in the non-pregnant group (r_s_ = 0.367, *p* = 0.0120). 

*Follicle-Stimulating Hormone (FSH):* Previous research indicated that basal, day 3 (D3) FSH values exceeding 10 IU/L could signify a poor response to ovarian stimulation and predict unfavorable IVF outcomes [37]. However, recent studies questioned its usefulness as a screening test for subfertility. Nowadays, more informative biomarkers, such as Antral Follicle Count (AFC) and anti-Müllerian hormone (AMH), are used to predict ovarian response and reserve capacity [38]. In our study, we detected higher intrafollicular Gal-9 levels in women with FSH levels exceeding 10 IU/L (Figure 4A). When we divided the samples into two FSH subgroups (<10 IU/L and >10 IU/L) and analyzed them according to the IVF outcomes, patients who did not conceive showed higher Gal-9 values compared to women who achieved pregnancy following embryo transfer. Nonetheless, these differences were not statistically significant in either group (Figure 4B).

*Luteinizing Hormone (LH):* While it is widely recognized that elevated D3 LH levels are often associated with polycystic ovary syndrome [39], the impact of altered LH production on female reproduction remains a subject of controversy. In our study we observed increased Gal-9 levels in the group with >3 IU/mL LH levels and found a significant, weak positive correlation between LH and Gal-9 values (r_s_ = 0.2240, *p* = 0.0476) (Figure 4C,D).

*FSH/LH ratio:* It was shown that an elevated basal FSH/LH ratio is linked with inferior IVF outcomes and higher cancellation rates [40]. Therefore, we examined the connection between high (>2) or low (<2) FSH/LH ratios and FF Gal-9 levels, but no significant difference was found. When considering the IVF outcomes, significantly higher Gal-9 values were detected in the non-pregnant group with <2 FSH/LH ratio (*p* = 0.0188) (Figure 4E).

### 2.4. Association with Treatment-Related Data and Clinical Parameters

*Total FSH dose/day:* It is generally accepted that controlled ovarian hyperstimulation (COH) with higher total doses of gonadotropins results in a higher yield of collectible mature oocytes—especially in younger women with good ovarian reserve [41]. However, a recent retrospective study showed its negative impact on the live birth rate following fresh embryo transfer [42]. This adverse effect persisted in women of advanced age (>35 years), even among those with a good prognosis for IVF (BMI < 30 kg/m2, no ovulatory dysfunction). In our study, the FF Gal-9 levels did not differ significantly between women undergoing <2500 IU/day or >2500 IU/day FSH dose stimulation; however, its level was higher in the non-pregnant cohort and the high total dose FSH subgroups (Figure 5A,B). 

*Body mass index (BMI):* Abnormal BMI is frequently linked to hormonal imbalances, anovulation, compromised oocyte quality, and adverse endometrium receptivity [43]. Women with BMI ≥ 25 kg/m^2^ face lower pregnancy rates and increased risk of complications. Therefore, we examined the relationship between the calculated BMI of our infertile patients and their Gal-9 levels. While higher BMI is associated with increased soluble lectin content, no significant relationship was found between the tested variables (Figure 5C).

### 2.5. Interactive Effect of Multiple Clinical Parameters

Next, a comprehensive analysis using bivariate correlation and multivariate regression was used to predict the combined effect of different fertility-related clinical parameters (including BMI, peak E2, basal FSH, total FSH dose/day, basal LH, infertility duration, and the total number of IVF cycles) on FF Gal-9 levels within the same model. The analysis showed a weak fit for the data (F = 1.605, *p* = 0.148) with an R^2^ = 0.137 (*n* = 71), indicating that tested variables could explain 13.7% of intrafollicular Gal-9 values. Only the total daily FSH dose showed a weak but significant negative correlation with follicular Gal-9 levels in both models (r= −0.247, *p*= 0.022, B= −0.030, *p* = 0.021). All other parameters demonstrated independent associations with the soluble Gal-9 values (Table 1).

### 2.6. Follicular and Serum Galectin-9 Levels in IVF Patients and Controls

The follicular fluid consists of soluble factors secreted locally by the follicular cells and elements transferred from the blood plasma, making it essential to identify the potential sources of follicular soluble Gal-9. To investigate this context, we collected serum and FF specimens from IVF patients on the day of oocyte recovery alongside sera from fertile, age-matched women. We observed that following ovarian stimulation, the serum of our infertile patients contained a higher amount of soluble Gal-9 compared to their intrafollicular levels. Furthermore, both the follicular fluid aspirates and serum samples of IVF patients had significantly higher Gal-9 levels (*p* < 0.05) than that of the sera obtained from healthy, fertile controls (Figure 6A). These results suggest that controlled ovarian stimulation may affect the final composition of the follicular fluid by triggering intraovarian Gal-9 expression and promoting its transport from the circulation across the blood-follicle barrier (Figure 6B). However, further studies are needed to confirm whether these processes play a role in altering intrafollicular Gal-9 levels during oocyte development.

### 2.7. Diagnostic Value of Intrafollicular Gal-9 Measurement

Finally, the receiver operating characteristic curve analysis was used to determine the diagnostic value of soluble, follicular Gal-9 as a predictive, minimally invasive laboratory marker in the IVF program. (Figure 7).

*ROC-1:* We revealed a remarkable discrimination ability of Gal-9 in predicting fertilization success (*p* < 0.0001) with a calculated area under the curve (AUC) value of 0.848 [95% confidence interval (CI) 0.750–0.919]. The optimal cut-off point was >163.943 pg/mL with 69.33% sensitivity and 100% specificity. Considering the calculated 65/71% fertilization success rate based on the presence of pronucleus/embryo, the positive predictive value (PPV) reached 100%, and the negative predictive value (NPV) was 63.71/57.12%, with a diagnostic accuracy of 80.07/78.23%.

*ROC-2:* In the context of predicting IVF outcome, Gal-9 showed moderate value. The estimated AUC was 0.652 [95% CI 0.531–0.759] at a cut-off point of ≤213.649 pg/mL with 95.24% sensitivity and 40.38% specificity. According to the 27.5/23.61% pregnancy rate calculated by the success of IVF/presence of a fetal heartbeat, the PPV was 37.7/33.1%, and the NPV was 95.7/96.5% with a diagnostic accuracy of 55.47/53.34%, respectively. Though the overall performance was lower (*p* = 0.019), it still showed sufficient prognostic power.

*In comparison* to other follicular fluid biomarkers, Gal-9 demonstrates superior diagnostic potential for predicting fertilization competence of the oocyte compared to AMH [44], prolactin [45], dihydrotestosterone [46], and soluble HLA-G [47], although its diagnostic value remains less effective than IL-10 [48]. In terms of predicting successful pregnancy, Gal-9 shows a lower predictive value than IL-8 [49] and leptin [50], but outperforms E2, IGF1, and lactoferrin [51,52,53]. These comparative findings are summarized in Table 2.

## 3. Discussion

In the present study, we characterized the tissue- and cell-specific expression of Gal-9 in the ovary and evaluated its potential clinical usefulness as a minimally invasive biomarker in the IVF program. While several associations between FF Gal-9 levels and clinical parameters showed interesting trends, many did not reach statistical significance. Therefore, our findings should be considered exploratory, and larger, more comprehensive multicenter studies are needed to validate them in the future.

In this study, intracellular Gal-9 expression was confirmed by immunohistochemistry in ovarian epithelial cells, stromal cells, and within the ovarian follicles during various stages of oogenesis. Moderate labeling was detected in the cytoplasm of developing oocytes and the plasma membrane of the mature ovum. We observed that Gal-9 expression was transiently increased from the primordial to the primary follicle stage, followed by a decline in mature oocytes. Given the established involvement of Gal-9 in regulating cell proliferation and apoptosis [16,54], we hypothesize that it may influence oocyte fate decisions, potentially promoting either follicular growth or atresia in response to local regulatory signals. The prominent labelling of the plasma membrane in mature oocytes—a region critical for sperm binding and fusion—raises the possibility that Gal-9 may also participate in fertilization. Furthermore, the nuclear Gal-9 labelling in subsets of theca interna, theca externa, and cumulus granulosa cells may suggest that intracellular Gal-9 may directly modulate the biological functions of these cells, potentially influencing follicular fluid production and oocyte development. While these preliminary observations indicate a possible involvement of Gal-9 in oogenesis, direct functional evidence remains lacking, and further studies are required to elucidate the specific regulatory role of Gal-9 in follicular development. Follicular fluid, which is essential for oocyte development, is derived from both follicular granulosa cells and systemic circulation, so alterations in their function or blood composition can modify the amount of locally produced or secreted substances within the follicular fluid. Previous research by Popovici et al. [21] has indeed shown that Gal-9 expression in the endometrium varies with the menstrual cycle (particularly during the mid- and late-secretory phases of the cycle) and during the implantation window, suggesting a hormonal influence on Gal-9 production in the female reproductive system. In agreement with this observation, we found significantly increased Gal-9 levels in the serum and follicular fluid of IVF patients after ovarian stimulation in comparison to fertile women. These results raise the possibility that, besides local production, Gal-9 may enter the follicular fluid from the bloodstream via the blood-follicle barrier, potentially affecting oocyte development and fertilization competence of the egg. However, further investigations are required to identify the precise source of soluble Gal-9 in the follicular fluid.

In addition to investigating ovarian Gal-9 expression, we analyzed IVF-related clinical background data of infertile patients to assess the diagnostic and prognostic value of follicular Gal-9 as a minimally invasive laboratory marker in the IVF program. We verified that the soluble form of Gal-9 is present in the follicular fluid, and its concentration fluctuates from 101.229 pg/mL to 446.005 pg/mL with the number of mature oocytes. A previous comprehensive survey involving 400,135 IVF cycles [55] revealed a strong association between the live birth rate and the number of retrieved eggs per fresh IVF cycle. They identified that the likelihood of success increased from 1 to 15 collected oocytes, plateaued between 15 and 20 eggs, and declined beyond 20. In our study, the highest Gal-9 levels were measured in the low-responder women with <3 oocytes retrieved, while the lowest lectin values were detected in the high-responder patients with >15 oocytes retrieved after COH. Furthermore, within the subgroup of high responders, markedly elevated Gal-9 levels were significantly associated with those women who subsequently developed OHSS. These findings suggest a potential link between FF Gal-9 levels and ovarian response. However, due to the small sample size, the possibility that Gal-9 could serve as a predictor to identify patients at risk for OHSS warrants further investigation, and larger patient cohorts are necessary to validate these preliminary findings in the IVF program.

Intriguing findings emerged after examining the relationship between follicular Gal-9 levels and infertility-related clinical parameters. Higher Gal-9 levels were detected in women with infertility duration exceeding 5 years or elevated BMI, both factors associated with reduced chances of spontaneous pregnancy and adverse outcomes [35,55]. Further analysis revealed stimulation cycle-dependent variations of follicular lectin levels depending on IVF outcomes. Due to lack of statistical support, these findings are exploratory, and further studies are needed to explore the underlying mechanisms of these changes.

Although Gal-9 is known to participate in various cellular and molecular interactions [55], there is currently no published evidence directly linking Gal-9 to hormonal regulators of oogenesis. In our study, we observed that women with elevated basal serum FSH and LH levels—who either experienced failed IVF cycles or required higher total daily FSH doses during stimulation—tended to exhibit increased FF Gal-9 levels compared to those with normal hormonal profiles and successful IVF outcomes. While these findings are observational, we identified a statistically significant, albeit weak, positive correlation between day 3 serum LH and FF Gal-9 levels, suggesting a potential interaction between these molecules. Additionally, Hu et al. [56] demonstrated that stimulation of sexually immature rabbits (*Oryctolagus cuniculus*) with pregnant mare serum gonadotropin (PMSG) led to the downregulation of ovarian microRNA-34 (miR-34), which is predicted to target *LGALS9* the gene encoding Gal-9. These findings raise the possibility of both hormonal and transcriptomic regulation of Gal-9 expression within the ovary. However, the associations observed in our study remain preliminary and should be interpreted with caution. Further mechanistic studies are needed to elucidate the relationship between gonadotropins and Gal-9 during oogenesis.

Our following interesting observation was the significantly elevated Gal-9 levels in women with elevated (>3000 pmol/L) serum peak E2. Additionally, there was a weak, positive correlation in those women who failed to achieve pregnancy. The potential link between these variables is further supported by our earlier data indicating menstrual cycle-dependent fluctuations of serum Gal-9 in healthy women. Specifically, we found that soluble Gal-9 levels in the serum of fertile women with regular menstruation cycles typically range from 319 to 393 pg/mL during the follicular phase, peak around ovulation at 1950 pg/mL, remain elevated during the early luteal phase, and then decline to 294 pg/mL during the late luteal phase preceding menstruation. Furthermore, we detected significantly elevated Gal-9 expression in endometriosis, a condition primarily dependent on E2 [33]. Other studies also support the potential interaction between Gal-9 and E2. In vitro cell culture results by Murata et al. [57] demonstrated significantly increased *LGALS9* mRNA expression following 12-day treatment with E2 and medroxyprogesterone acetate. Furthermore, in silico gene expression data of Naciff et al. [58] predicted an interaction between Gal-9 and E2 in rats. Additionally, Anifandis et al. [59] reported a similar, coordinated interaction between elevated E2 and increased serum and follicular leptin levels, which resulted in poor oocyte yield, low implantation success, and decreased pregnancy rates after IVF. Further studies are required to confirm the possible link between these factors in the regulation of oocyte development.

In clinical practice oocyte maturity is typically assessed using light microscopy, which provides only a limited and superficial evaluation of cumulus cell morphology or the denuded oocyte prior to IVF. Ovarian stimulation may further complicate this assessment by bypassing the natural selection mechanisms, increasing the likelihood of retrieving oocytes with suboptimal developmental potential and a higher risk of fertilization or implantation failure [60]. Consequently, there is a growing demand for laboratory markers that reliably reflect oocyte quality and accurately predict its developmental competence. In this context, we investigated the diagnostic potential of follicular Gal-9 as a non-invasive marker in the IVF cycles. We performed ROC curve analysis, which revealed that soluble Gal-9 possesses strong predictive value for fertilization success, with an AUC of 0.848 (95% CI: 0.750–0.919; *p* < 0.0001), achieving 63.3% sensitivity and 100% specificity. This performance exceeded that of AMH and several established follicular markers, including adipokines (e.g., prolactin), DTH, and sHLA-G, and was also superior to age, menstrual cycle length, basal FSH, E2, and BMI, suggesting that Gal-9 may represent a more reliable predictor of oocyte’s fertilization potential than most of the known biomarkers. We propose that this promising diagnostic performance may support embryologists in identifying the highest-quality, or so-called “golden,” oocytes in future IVF applications. Additionally, follicular Gal-9 exhibited a moderate, yet statistically significant, predictive value for overall IVF outcome (AUC = 0.652; 95% CI: 0.531–0.759; *p* = 0.019). Although this performance was weaker than that of IL-8 and leptin, it exceeded the predictive value of follicular E2, IGF-1, and lactoferrin.

Based on our findings, it appears plausible that optimal oocyte maturation requires follicular fluid Gal-9 levels above a critical threshold (163.943 pg/mL), and that both abnormally low and excessively high concentrations may be associated with impaired oocyte maturation and reduced fertilization rates. Although these results are encouraging, further clinical studies involving larger cohorts are essential to validate the diagnostic utility of Gal-9 in IVF treatments. Given the multifactorial nature of follicular fluid composition and the complexity of oocyte maturation, it is unlikely that Gal-9 alone can fully account for fertilization competence. Nonetheless, from both clinical and diagnostic perspectives, we propose that integrating Gal-9 into a broader biomarker panel—alongside established IVF indicators—could improve the specificity, sensitivity, and overall reliability of preimplantation markers in ART.

The general overview of our study design and the main findings are visually summarized in a Graphical abstract created with BioRender.com (https://www.biorender.com (accessed on 7 July 2025)).

## 4. Materials and Methods

### 4.1. Patients and Samples

Our study was conducted between 2018 and 2022 at the University of Pecs, involving 130 clinical samples. Before conducting the research, informed consent was signed by each participant. Figure A1 shows the inclusion and exclusion criteria of study participants. Their main demographic, clinical, and laboratory data are summarized in Table A1 and Table A2.

During the study period, 87 FF was collected, and 83 FF was investigated from randomly chosen infertile women using transvaginal ultrasound-guided aspiration as part of the standard IVF procedure. In addition, we included serum-follicle fluid pairs of 19 infertile women and sera of 16 fertile volunteers for comparative analysis. Collection of the specimens, acquisition of clinical data, and the execution of the IVF process were carried out in the Assisted Reproduction Unit of the Department of Obstetrics and Gynecology. The research work was performed in the Department of Medical Microbiology and Immunology. The selection of infertile couples was carried out consecutively, and their enrollment in the IVF program was approved by two independent gynecologists. Importantly, none of the patients had a history of metabolic or vascular disorders. Before initiating the IVF process, all couples underwent clinical examinations and comprehensive laboratory tests. Serum hormone profiles were assessed on the 3rd and 21st days of the unstimulated cycles. To decrease pre-trial bias, different examiners were involved in the recruitment of IVF patients and controls and clinical data collection. Blinded scientists performed laboratory experiments and analyzed the research data.

### 4.2. Controlled Ovarian Hyperstimulation (COH)

The study population consisted of patients with various primary infertility diagnoses, including anatomical reasons (18/83), endocrine disorders (10/83), advancing age (6/83), unsuccessful previous artificial homolog insemination (24/83), male factors (9/83), unexplained infertility (9/83), and mixed etiology (7/83). To achieve ovarian stimulation, the gonadotropin-releasing hormone agonist triptorelin (Gonapeptyl^®^; Ferring, Kiel, Germany) was used to synchronize follicles using either long or short protocols. Depending on follicular maturation, stimulation was started with individual dosages (150–250 IU/day) of recombinant FSH (Gonal-F^®^; Serono, Aubonne, Switzerland). The starting dose was adjusted for BMI and age. The daily FSH dose was increased to 300 IU for those patients who previously were low responders. Follicular maturation was monitored with transvaginal ultrasonography from the 6th day of the cycle and subsequently on alternate days. The gonadotropin dosage was adjusted individually according to the size of the follicles. Once at least two follicles exceeded 17 mm in diameter, 250 μg of recombinant human choriogonadotropin alfa (hCG) (Ovitrelle^®^; Merck KGaA, Merck Serono, Damstadt, Germany) was added to induce final oocyte maturation. Oocytes and FF samples were collected 36 h later by transvaginal ultrasound-guided puncture under routine intravenous (IV) sedation using a SonoAce 6000C 2D real-time ultrasound scanner (Samsung Medison Co., Ltd., Sonarmed, Budapest, Hungary) fitted with a 4–8 MHz endovaginal transducer.

### 4.3. Collection of Biological Samples

Follicular fluids: samples were aseptically collected by antral follicle puncture into sterile tubes containing G-MOPS^TM^ medium (Vitrolife, Goteborg, Sweden). The follicular fluid aspirates were centrifuged at 252 g for 10 min (min). The resulting supernatants were aliquoted and stored at −80 °C to ensure optimal preservation until analysis.

Serum samples: native peripheral blood samples were obtained from 19 infertile women on the morning of follicular puncture before sedation. Besides, 16 age-matched fertile donors were used as controls. Following venipuncture, the samples were allowed to coagulate and centrifuged for 10 min at 3000 rpm to separate cellular components. The serum fraction was decanted, aliquoted, and stored at −80 °C until analysis.

### 4.4. In Vitro Fertilization (IVF) Procedures

The fertilization process was carried out with conventional IVF (cIVF) or with intracytoplasmic sperm injection (ICSI), depending on the andrological status (total sperm count <15 million/mL), maternal age (>35 years), and the number of previous IVF cycles (>2). The oocytes selected for ICSI were denuded with hyaluronidase, and only the mature, metaphase-II (MII) oocytes that showed the presence of the first polar body were chosen for fertilization. ICSI was performed 3–6 h later in the G-MOPS^TM^ medium (Vitrolife, Goteborg, Sweden), and the remaining oocytes were fertilized with the cIVF method in a bicarbonate-buffered G-IVF^TM^ medium (Vitrolife, Goteborg, Sweden). Fertilization was evaluated 24 h later, and embryos were cultured individually in G-Iv5^TM^ medium (G1, Vitrolife, Goteborg, Sweden) under oil. From day 3 to the blastocyst stage, embryos were cultivated under oil in 40 μL G-21Mv5 medium (G2, Vitrolife, Goteborg, Sweden).

Embryo transfer (ET) was performed three to five days after fertilization. Following the embryo scoring system consensus of the European Society of Human Reproduction and Embryology (ESHRE), only the highest-quality embryos were transferred. At the patient’s request, one to three embryos were transferred, and the remainder were frozen in compliance with Hungarian law. Clinical background data were collected only from patients undergoing fresh embryo transfer. No patients who underwent frozen-thawed transfer cycles were included in the study. After fresh ET, a total of 300 μg oral progesterone (Utrogestan; Lab. Besins International S.A., Paris, France) was prescribed to the patients to support the luteal phase. Implantation was confirmed 21–28 days following embryo transfer (ET) by detecting a gestational sac with a transvaginal ultrasound exam. The presence of a fetal heartbeat verified clinical pregnancy. Implantation failure was defined by the lack of the former and the absence of β-hCG production on week 2 after ET.

### 4.5. Immunohistochemistry (IHC)

Intracellular Gal-9 expression was assessed on 12 representative 4–5 µm formalin-fixed paraffin-embedded histologic sections [ovary (n = 11), endometrium (n = 1)] that were provided by the Department of Pathology, University of Pecs. After deparaffinization and rehydration, antigen retrieval was carried out using modified citrate buffer (pH = 6, DAKO, Glosup, Denmark) in a 95 °C water bath for 30 min. Endogenous peroxidase activity was blocked with 3% H_2_O_2_. After 3 × 5 min washing with Tris Base Buffer + 0.05% Tween (pH = 7.4) (TBST), nonspecific binding sites were blocked with 3% Bovine Serum Albumin (BSA, Merck-Life Science, Budapest, Hungary) for 20 min. Then, monoclonal anti-human Gal-9 antibody (clone 1G3, Merck KGaA, Darmstadt, Germany) was applied at a dilution of 1:500 in TBST+1% BSA and incubated overnight at 4 °C in a humidified chamber. Negative controls were prepared by using isotype-specific mouse IgG1κ instead of primary antibody. After washing, polyclonal goat anti-mouse immunoglobulins-HRP (DAKO, Glosup, Denmark) was used at a dilution of 1:100 and incubated for 30 min at room temperature (RT). Negative controls were prepared by using isotype-specific nonspecific mouse IgG1α instead of primary antibody. Following 3 × 5 min washing, the immunostaining was visualized with the DAB+ Substrate Chromogen System (DAKO, Glosup, Denmark). Counter-staining was performed with Mayer’s Hematoxylin (Merck KGaA, Darmstadt, Germany), followed by mounting with Faramount Aqueous Mounting Medium (DAKO, Glosup, Denmark). The slides were scanned using a PANNORAMIC Midi Digital Slide Scanner (3D HISTECH Ltd., Budapest, Hungary) and analyzed with the CaseViewer v.2.4 software.

### 4.6. Enzyme-Linked Immunosorbent Assay (ELISA)

The concentration of soluble Gal-9 was determined with a commercial, competitive-type ELISA (#E01G0073, BlueGene Biotech Co., Ltd., Shanghai, China) by strictly following the manufacturer’s instructions. Briefly, 100 μL/well of pre-diluted recombinant Gal-9 standards (range 0–1000 pg/mL) and 100 μL/well of clinical samples were pipetted to the appropriate test wells of the ELISA plate pre-coated with anti-Gal-9 antibody. Subsequently, 10 μL/well of balance solution was added to the FF specimens. To decrease intra- and inter-assay variances, the samples were examined in duplicate. Next, 50 μL of Gal-9-HRP conjugate was pipetted to the wells except for the BLANK, and the plate was incubated at 37 °C for one hour. After five washes with 350 μL of wash buffer, 50–50 μL of Substrate A and B were pipetted to each well and incubated for 25 min at 37 °C in the dark. Finally, 50 μL/well of stop solution was added to terminate the enzyme reaction. The optical density of the wells was measured at 450 nm using a SPECTROstar Nano Microplate Reader (BMG Labtech, Ortenberg, Germany). Gal-9 concentrations were determined by the MARS Data Analysis Software ver. 3.32 (BMG Labtech, Ortenberg, Germany) and presented as pg/mL. The sensitivity of the assay was 1 pg/mL.

### 4.7. Statistical Analysis

Statistical analyses were executed with MedCalc v.16.8 (Medcalc Software Ltd., Ostend, Belgium, https://www.medcalc.org) and GraphPad Prism v.3.00 for Windows (GraphPad Software, San Diego, CA, USA; https://www.graphpad.com) software packages. Multiple linear regression analysis was performed with SPSS Statistics v.22.0 software for Windows (IBM Corp., Armonk, NY, USA, https://www.ibm.com/products/spss-statistics).

First, descriptive statistics were accomplished, and the Shapiro-Wilk test was used to establish the distribution of the obtained data sets. To overcome outlier bias, the Tukey method was used to identify mild (far) and extreme (far out) outliers in each cohort. Outliers were excluded from further statistical analysis. Both the descriptive statistics and outlier analysis were repeated when new test groups were compared. Depending on the variance and distribution of the data sets, the values of two study groups were compared with unpaired t-tests without/with Welch correction or the Mann–Whitney U-test. The one-way ANOVA test with Tukey post hoc and the Kruskal–Wallis H test with Dunn’s comparison were applied for multiple comparisons. Spearman’s rho or Pearson correlation analysis examined the relationship between various clinical parameters and soluble Gal-9 levels. Fisher’s exact test was used to assess differences between the categorical data of the tested groups. Based on the correlation coefficient (the r_s_ or r-value), a positive (0 < r < 1) or negative (−1 < r < 0) relation was determined with different strengths. Multivariate linear regression analysis was performed to assess the impact of multiple predictors on Gal-9 levels in the same model. After calculating the coefficients of determination (R^2^ and adjusted R^2^), the ANOVA test was used to determine how the overall regression model fit the data. The differences were considered statistically significant if the calculated *p*-values were ≤0.05. The Receiver Operating Characteristic (ROC) curve was used to evaluate the diagnostic value of Gal-9 in the IVF program. The area under the curve (AUC) was considered significantly different from the null hypothesis if its value was >0.5.

## 5. Conclusions

In summary, to our knowledge, this study is one of the first to characterize the biological role and clinical usefulness of Gal-9 in the IVF program. Our findings suggest that Gal-9 may play an important role in oogenesis, and abnormally low or highly elevated Gal-9 levels potentially impair fertility, ovarian response, and embryo quality. Thus, Gal-9 may be one of several factors influencing IVF outcomes. Importantly, our data highlight the diagnostic potential of Gal-9 as a non-invasive predictor of fertilization success prior to embryo transfer. While Gal-9 alone may not fully capture fertilization competence, its integration into a multi-marker panel alongside established IVF indicators may improve the specificity, sensitivity, and overall reliability of preimplantation assessment. Clinically, these findings may offer insight into a complex mechanism of oocyte development, and if confirmed, could support the individualization of COH protocols to improve the cost-effectiveness of IVF.

Our study, however, has limitations: (1) Our study was based on clinical samples collected from a single IVF center. (2) Pooled FF samples were collected from the IVF patients; however, individual aspiration specimens would provide more precise insights into the relationship between Gal-9 and the tested clinical parameters. Nonetheless, obtaining individual follicle aspirations is challenging in practice due to the need for multiple punctures, which can increase the risk of bleeding and patient discomfort during the procedure. (3) Gal-9 expression was solely investigated through immunohistology analysis without further characterization at the mRNA level. (4) The fourth limitation is the absence of correlation data between intrafollicular Gal-9 and other well-established and commonly used IVF predictors, such as AFC or AMH. In the future, we plan to expand our research to expression and functional studies that can identify pathologies associated with abnormal Gal-9 levels in the female reproductive system.

## Figures and Tables

**Figure 1 ijms-26-07672-f001:**
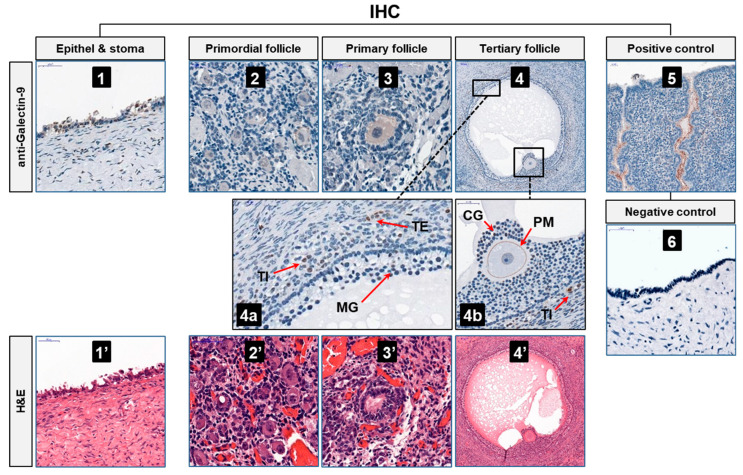
**Galectin-9 expression in human tissue sections after cycle induction.** Gal-9 immunohistochemistry (upper line, images ((**1**)–(**5**))) and hematoxylin-eosin staining (lower line, images ((**1′**)–(**4′**))) of human ovary and endometrium sections. Upper line: representative images of Gal-9 positive ovarian epithelial and stroma cells (**1**), primordial follicles (**2**), primary follicles (**3**), and tertiary follicles (**4**). Middle line, image (**4a**): red arrows mark nuclear positive theca interna, theca externa, and mural granulosa cells. Image (**4b**): red arrows indicate some weak nuclear positive cumulus granulosa, strong positive theca interna cells, and Gal-9 labeling of the plasma membrane of the mature oocyte. Sections were counterstained with hematoxylin and eosin. Positive control: secretory phase endometrium (upper line, image (**5**)). Negative control: ovary section stained with an isotype-matched non-specific antibody (middle line, image (**6**)). Magnifications (scale bars): 40× (50 μm), except image-4: 10× (100 μm), image-5: 20× (100 μm). Abbreviations: IHC = immunohistochemistry, H&E = hematoxylin and eosin, theca interna = TI, theca externa = TE, mural granulosa = MG, cumulus granulosa = CG, plasma membrane = PM.

**Figure 2 ijms-26-07672-f002:**
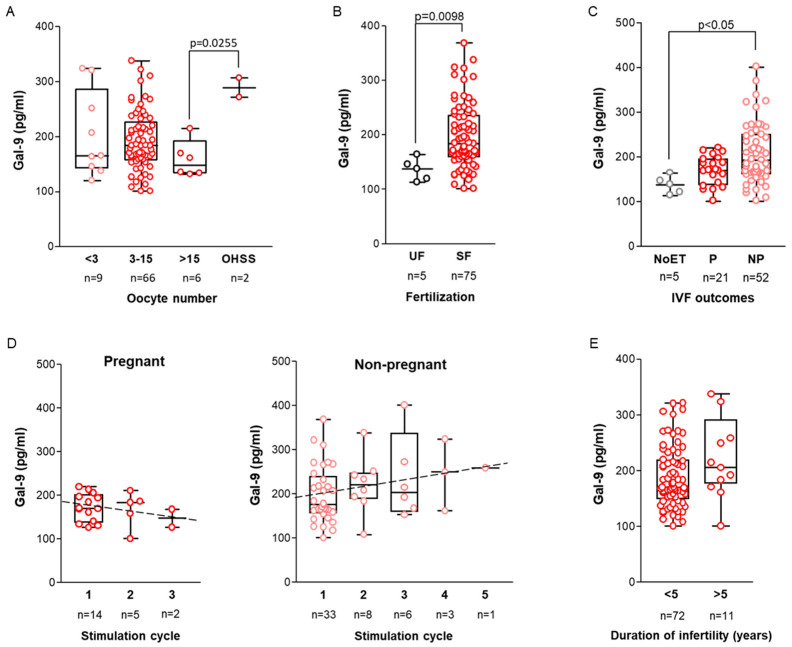
**Association of follicular fluid Galectin-9 levels with IVF-related parameters.** (**A**) Relationship between intrafollicular Gal-9 levels and the total number of oocytes retrieved per patient. (**B**) Association between follicular Gal-9 levels and fertilization success (**C**) and its relationship with IVF outcomes. (**D**) Alteration in FF Gal-9 levels during repetitive IVF cycles in pregnant and non-pregnant groups. The X-axis marks the number of total IVF cycles. Trends are indicated with intermittent lines. (**E**) Impact of infertility duration on Gal-9 levels. Scattered boxplots: boxes show the interquartile ranges, and whiskers indicate the most extreme observations. The middle lines mark median values. Individual values are shown as circles. Abbreviations: OHSS = ovarian hyperstimulation syndrome, UF = unsuccessful fertilization, SF = successful fertilization, IVF = in vitro fertilization, NoET = No embryo transfer, P = pregnant, NP = non-pregnant.

**Figure 3 ijms-26-07672-f003:**
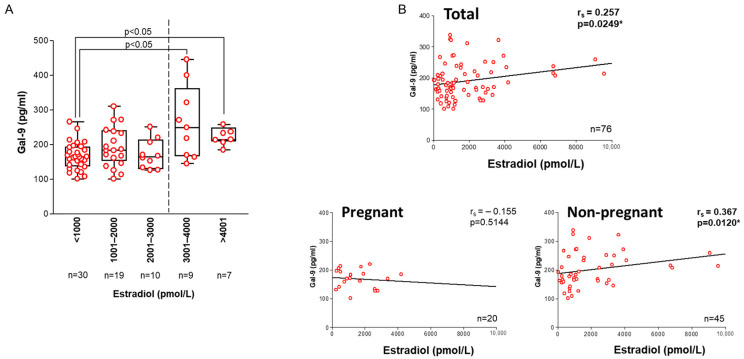
**Association of follicular Galectin-9 values with peak serum estradiol levels.** (**A**) Relationship between Gal-9 and peak serum estradiol levels in IVF patients. The dashed line separates samples with peak serum E2 levels above and below 3000 pg/mL. (**B**) Correlation between Gal-9 and peak serum estradiol levels in all IVF patients (total) and the pregnant and non-pregnant cohorts. Scattered boxplots: boxes show the interquartile ranges, and whiskers indicate the most extreme observations. The middle lines mark median values. Individual values are shown as circles. The asterisk (*) indicates statistically significant values.

**Figure 4 ijms-26-07672-f004:**
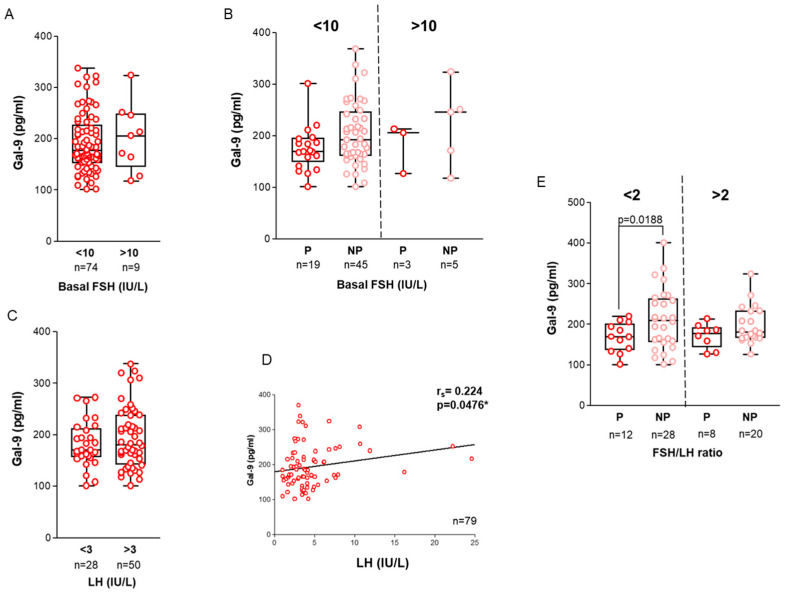
**Association of follicular Galectin-9 values with basal serum FSH and LH levels.** (**A**) Association between Gal-9 and basal FSH levels; (**B**) and its relation to the IVF outcome in the <10 IU/L and >10 IU/L FSH cohorts. The X-axis indicates the IVF outcomes. (**C**) Relationship between Gal-9 and serum LH levels (**D**) and their correlation in the examined samples. (**E**) Association between Gal-9 and FSH/LH ratios and its relation to IVF outcomes in the <2 and >2 FSH/LH cohorts, separated by a dashed line. The X-axis indicates the IVF outcomes. Scattered boxplots: boxes show the interquartile ranges, and whiskers indicate the most extreme observations. The middle lines represent median values. The asterisk (*) indicates statistically significant values. Individual values are presented as circles. Abbreviations: FSH = follicle-stimulating hormone, LH = luteinizing hormone, IU = international unit.

**Figure 5 ijms-26-07672-f005:**
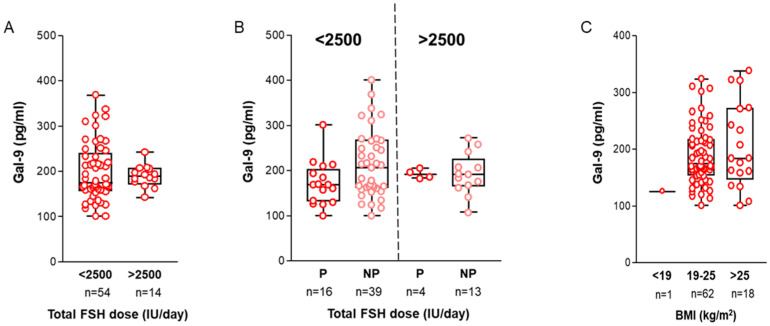
**Association of follicular Galectin-9 levels with clinical parameters.** (**A**) Relationship between Gal-9 levels and total daily FSH dose used for controlled ovarian hyperstimulation. (**B**) Soluble Gal-9 values in the FF of IVF women undergoing <2500 IU/day and >2500 IU/day total FSH-dose stimulation, separated by a dashed line. The X-axis indicates the outcome of IVF. (**C**) Relationship between FF Gal-9 levels and calculated BMI values (kg/m^2^) of the IVF patients. Abbreviations: FSH = follicle-stimulating hormone, IU = international unit, BMI = body mass index.

**Figure 6 ijms-26-07672-f006:**
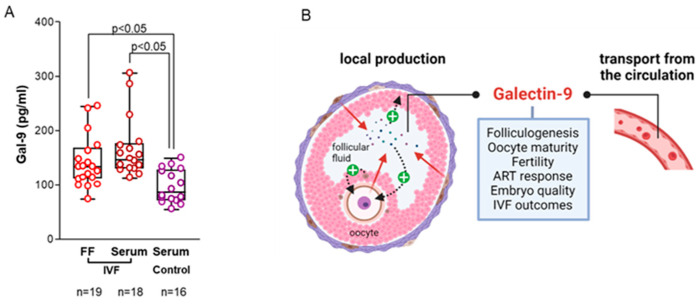
**Follicular and serum Galectin-9 levels in infertile patients and health controls.** (**A**) Comparative analysis of follicular and serum Gal-9 levels of IVF patients and fertile control women. Scattered boxplots: boxes show the interquartile ranges, and whiskers indicate the most extreme observations. The middle lines represent median values. Individual values are presented as circles. (**B**) Schematic diagram showing the potential sources and biological function of Gal-9 in the follicular fluid. The illustration was created with BioRender software. Abbreviations: FF = follicular fluid, IVF = in vitro fertilization, ART = assisted reproductive technology.

**Figure 7 ijms-26-07672-f007:**
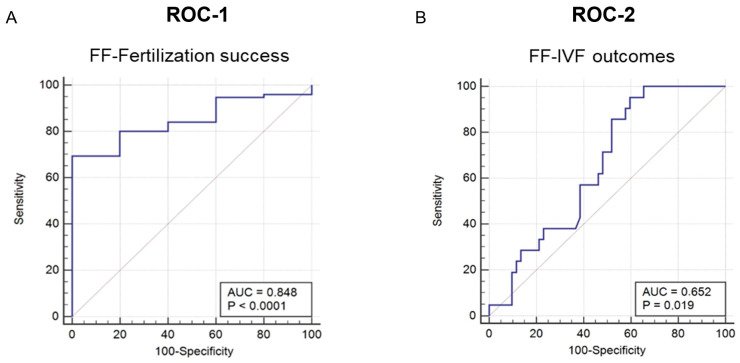
**ROC analysis for the values of Gal-9 in the follicular fluid.** (**A**) ROC-1 analysis using variables of fertility success and follicular Gal-9 levels (n = 80). (**B**) ROC-2 analysis using variables of IVF outcome and follicular Gal-9 levels (n = 73). AUC and *p*-values are indicated in boxes.

**Table 1 ijms-26-07672-t001:** Bivariate correlation and multivariate regression analysis between follicular Gal-9 levels and fertility-associated factors (n = 71).

Variables	Bivariate Correlation		Multiple Regression		
			R^2^ = 0.137, *p* = 0.148		
	r	*p*	B	SE	*p*
(Constant)	-	-	153.666	65.500	**0.022 ***
BMI	0.056	0.613	2.872	2.417	0.239
E2	0.201	0.072	0.003	0.002	0.216
FSH	−0.011	0.916	2.441	3.194	0.447
**Total FSH dose/day**	**−0.247**	**0.022 ***	**−0.030**	**0.013**	**0.021 ***
LH	0.151	0.173	2.845	2.866	0.324
Infertility duration	0.005	0.963	−4.183	5.284	0.434
IVF cycles	0.048	0.657	9.727	9.509	0.310

Abbreviations: BMI = body mass index, E2 = estradiol, FSH = follicle stimulation hormone, LH = luteinizing hormone, IVF = in vitro fertilization, r = Pearson’s correlation coefficient, B = estimated model coefficient, SE = standard error of the mean. The differences were considered significant when the *p*-values were ≤0.05. Significant values are shown in bold, and significant p-values denoted by an asterisk (*).

**Table 2 ijms-26-07672-t002:** Diagnostic performance of Gal-9 and different follicular fluid biomarkers in infertility.

Fertilization	AUC	SENS	SPEC	IVF Outcome	AUC	SENS	SPEC
**Gal-9**	**0.848**	**63.3**	**100**	**Gal-9**	**0.652**	**95.2**	**40.4**
AMH	0.813	98.4	62.5	E2	0.283	27.3	37.5
Prolactin	0.760	77.0	78.0	IGF1	0.629	87.9	40.0
DHT	0.756	81.7	65.4	Leptin	0.856	80.8	77.3
IL-10	0.910	87.0	74.0	IL-8	0.920	100.0	78.6
sHLA-G	0.676	71.4	75.6	Lactoferrin	0.513	45.0	65.1

Abbreviations: AUC = Area Under the Curve, SENS = sensitivity, SPEC = specificity, IVF = in vitro fertilization, AMH = anti-Mullerian hormone, DHT = dihydrotestosterone, IL = interleukin, sHLA-G = soluble human leukocyte antigen-G, E2 = estradiol, IGF1 = insulin-like growth factor-1.

## Data Availability

The data underlying this paper are available in the submitted article. The subjects in this study have not concomitantly been involved in other trials. Data regarding any of the subjects in the study have not been previously published unless specified. Data will be made available to the editors for review or query upon request.

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
