# Peer review of "The Sweet Side of IVF: Biological Role and Diagnostic Potential of Galectin-9 in Female Infertility"

_ijms, 2025, doi:10.3390/ijms26167672_

Round 1
Reviewer 1 Report
Comments and Suggestions for Authors
Dear authors,
I reviewed your manuscript entitled The sweet side of IVF: Biological role and Diagnostic Potential of Galectin-9 in Female Infertility submitted to the International Journal of Molecular Science with great interest.
This study provide a meaningful insights into the diagnostic role of Galectin-9 in predicting the IVF success.
In this experimental study the authors have measured the concentrations of Galectin-9 in the serum and FF from infertile women undergoing IVF procedures and their healthy counterparts as control. They also measured the expression of Gal-9 in the ovary in a spatio-temporal manner.
They found that FF levels of Galectin-9 play a critical role in oogenesis ,and elevated concentrations impairs ovarian response and embryo competency that results in infertility.
Considering the limitation of the study mentioned by by authors, this study can add constructive information to the field.
There are minor concerns that needs to be considered by authors.
There are some tempos that need to be corrected.
lines 50-52 in the Introduction sections is not clear and needs to be rewritten.
The authors need to clarify that the transfer cycles are fresh or frozen thawed.
Also the number of blastocysts per transfer need to be clarified.
Regards
Reviewer 2 Report
Comments and Suggestions for Authors
This manuscript investigates the role of soluble Galectin-9 (Gal-9) in the follicular fluid of infertile women undergoing IVF, aiming to assess its potential as a diagnostic biomarker. The authors explore Gal-9 expression patterns in ovarian tissues, analyze its levels in various clinical contexts, and evaluate its predictive value for fertilization and IVF outcomes through ROC analysis.
The topic is timely and relevant to reproductive medicine, particularly in the search for non-invasive biomarkers to improve IVF prognosis. The manuscript is generally well written and clearly structured. The ROC-1 analysis shows promising diagnostic performance of Gal-9 in predicting fertilization success, which is a highlight of this study and could justify publication if appropriately emphasized and supported.
However, many of the statistical comparisons did not reach significance. While some trends are of interest, I'm not fully convinced that Gal-9 is a strong candidate biomarker based on the current data. The authors should clearly differentiate statistically supported results from exploratory observations, and avoid overinterpretation.
Given the limited statistical power and modest data strength across most analyses, I regret that I am unable to recommend the manuscript for publication in its current form.
Round 2
Reviewer 2 Report
Comments and Suggestions for Authors
The authors have addressed all my concerns. I recommend that the revised manuscript be published.